# RSM and BPNN Modeling in Incremental Sheet Forming Process for AA5052 Sheet: Multi-Objective Optimization Using Genetic Algorithm

**Xiao Xiao [1], Jin-Jae Kim [1], Myoung-Pyo Hong [1,2], Sen Yang [3] and Young-Suk Kim [4,\*]**

[1]  Graduate School, Kyungpook National University, Daegu 41566, Korea; xiaoye012512@knu.ac.kr (X.X.); jinjaekim@knu.ac.kr (J.-J.K.); mp77@kitech.re.kr (M.-P.H.)
[2]  Extreme Fabrication Technology Group, Korea Institute of Industrial Technology, Daegu 711-883, Korea
[3]  AAC Communication Technologies, Changzhou 213001, China; YangSen@aactechnologies.com
[4]  School of Mechanical Engineering, Kyungpook National University, Daegu 41566, Korea
[\*]  Correspondence: caekim@knu.ac.kr; Tel.: +82-053-950-5580

**Abstract:** In this study, the response surface method (RSM), back propagation neural network (BPNN), and genetic algorithm (GA) were used for modeling and multi-objective optimization of the forming parameters of AA5052 in incremental sheet forming (ISF). The optimization objectives were maximum forming angle and minimum thickness reduction whose values vary in response to changes in production process parameters, such as the tool diameter, step depth, tool feed rate, and tool spindle speed. A Box–Behnken experimental design was used to develop an RSM and BPNN model for modeling the variations in the forming angle and thickness reduction in response to variations in process parameters. Subsequently, the RSM model was used as the fitness function for multi-objective optimization of the ISF process using the GA. The results showed that RSM effectively modeled the forming angle and thickness reduction. Furthermore, the correlation coefficients of the experimental responses and BPNN predictions of the experiment results were good with the minimum value being 0.97936. The Pareto optimal solutions for maximum forming angle and minimum thickness reduction were obtained and reported. The optimized Pareto front produced by the GA can be a rational design guide for practical applications of AA5052 in the ISF process.

**Keywords:** incremental sheet forming; RSM; BP neural network; genetic algorithm; multi-objective optimization

## 1. Introduction

Incremental sheet forming (ISF) is a flexible sheet-forming process that has gained significant interest since the pioneering work of Iseki [1]. ISF is a highly localized deformation process in which a tool is programmed to move along a certain path to create the desired part geometry. A simple ISF process to manufacture a truncated cone is depicted in Figure 1 [2]. The workpiece/blank is clamped with a fixture. A pin-like tool is programmed to follow the circumference of a circle. After completing the first circle, the tool steps down and toward the center to start a new circular pass. After several passes, a truncated cone can be generated. Compared to the conventional press forming process, ISF can produce geometries of various parts directly from computer-aided design models and numerical control codes without complex tools or dies. Thus, this process not only saves energy, but also holds great potential for rapid prototyping of small quantities of parts. Additionally, it is known that ISF can significantly increase the formability of the sheet metal workpiece [3]. Furthermore, to enhance formability, several ISF schemes (micro ISF [4], robot-assisted ISF [5], and heat-assisted ISF [6]) have been proposed.

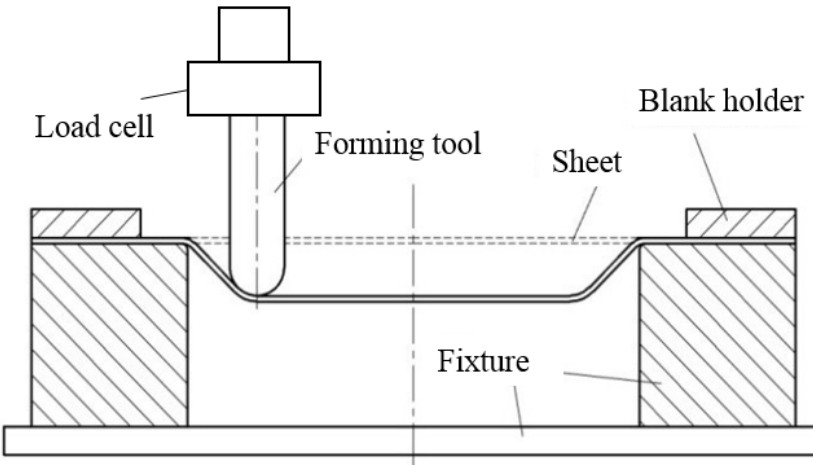

**Figure 1.** Single point incremental forming for truncated cone.

Recently, several papers that investigate the ISF processes through different approaches like analytical, numerical, and soft computing methods have been published. Behera [7] presented a review that describes the genesis and current state-of-the-art of the ISF process. Minutolo [8] researched the formability in the ISF process using the maximum forming angle. Fiorentino [9] carried out experimental investigations on formability of sheet metals in the single-point and two-point ISF processes considering different tool paths and incremental step height. Do [10] developed a new method and the associated apparatus to define the forming limit curve at fracture in the ISF process. The developed numerical simulation model showed good results to predict the fracture and thickness distribution. McAnulty [11] and Gatea [12] presented reviews of experimental results of the effects of different process parameters on the performance of ISF. Similarly, according to Leon [13] and Kim [14], the forming angle, tool diameter, vertical-step depth, horizontal-step depth, tool rotation speed (spindle speed), feed speed, sheet thickness, tool paths, and temperature are important forming parameters that affect the final forming result in the ISF process. In the ISF process, better formability is obtained when increasing the spindle speed and decreasing the feed rate of tool. At the same time, although decrease the size of the tool diameter can increase the formability, it will also increase the springback and surface roughness, resulting in a reduction of the accuracy of the formed part. Decreasing vertical step depth can improve formability and reduce springback, but it will significantly increase the forming time. On the other hand, tool paths optimization can reduce springback and enhance the thickness distribution of asymmetric parts. Therefore, the optimization of process parameters is required to achieve better formability and product quality in the ISF process. The finite element method (FEM) is today able to estimate the process parameter in the ISF process. However, the incremental nature of the ISF process requires a huge number of stages to model the entire process, implying large simulation times. Furthermore, several mechanical phenomena, like springback and bending, act during the process, thus requiring adequate constitutive models with many parameters that need to be calibrated [15]. On the other hand, various methods have been used in the past for the optimization of the parameters of the ISF process based on the experiment which can also give good analysis results. Angshuman [16] used grey relational analysis (GRA) to optimize the forming parameters for the ISF of AA5052 sheets. In their research, Taguchi's L9 orthogonal array, GRA, and analysis of variance (ANOVA) were used to achieve the optimum parameters for maximizing the formability and minimizing the roughness in the rolling, transverse, and angular direction. The results showed that lubrication has been identified as the highest contributing factor for all three directions, while the vertical step depth and speed were identified as the second and third contributing parameters, respectively, for both the rolling and the angular direction. Hani [17] presented an optimization of the two-point ISF process for AA1050 sheet using the response surface method (RSM). In their research, the Box–Bhenken experimental design (BBD) was utilized considering the mandrel angle,

tool diameter, sheet initial thickness and step depth as input parameters, and the thinning ratio and maximum resultant force as output responses. ANOVA was also performed to find the contribution of factors to the responses and it was inferred that all the regression models developed using the RSM were adequate for correlating the process factors and corresponding responses. It was also found that the wall angle was the most influential factor affecting thickness reduction, while the sheet thickness had the greatest influence on the axial force. Dakhli [18] proposed a method that combines two methods—Taguchi grey relational analysis (TG) and the RSM—in which the multi-response parameters of surface roughness, forming force, and manufacturing time are optimized by computing the grey relational grade. Based on the results, the material sheet and the lubricant were the most significant factors that affect the surface roughness, the forming forces, and the manufacturing time.

Artificial intelligence is widely used in various industries. Using artificial intelligence, not only is it possible to make good predictions but also optimize single or multiple objectives. The back propagation neural network (BPNN) is a machine learning tool that can be used to learn the relationships between the input and output variables to predict system performance. It works as a black box model that requires no detailed parameters of the system. The BPNN working principle was inspired by that of the human brain, and the network consists of inputs, several layers of neurons, and outputs. In its simplest form, an input is multiplied by weights, and then the product and a bias are summed up and sent into a transfer function to produce the final output. More recently, the artificial intelligence algorithms that train neural networks with the back propagation method have been applied to various problems in plasticity. Do [19] conducted research on the effect of hole lancing on the forming characteristic of ISF. In their study, the hole lancing on the blank shoulder near the forming area was designed and the BPNN algorithm was used to predict the springback in the ISF process. The results showed that hole lancing not only improved the formability considerably (the maximum forming angle increased from 60° to 64°), but also reduced profile error from 1.32 mm to 1.12 mm. Furthermore, the BPNN algorithm with the Levenberg–Marquardt approximation successfully predicted springback amount in the ISF with the average error of 4.052%. Simultaneously, Forcellese [20] presented multivariable empirical models based on artificial neural networks (ANN) to predict the flow and forming limit curves of the AZ31 magnesium alloy thin sheets. The results showed that the ANN captured the influence of temperature, strain rate, and fiber orientation on the flow curve shape, the stress values, and the effects of the process parameters on the forming limit curves without a priori knowledge of the complex microstructural mechanisms occurring during warm forming.

Genetic algorithms (GAs) are based on the principles of genetics found in nature. They are parallel and global search algorithms based on Darwin's theory of survival of the fittest [21]. GAs are an efficient comprehensive search method which automatically acquires and accumulates knowledge of the search space during the search process and has proper characteristics to control the search process to find the best solution. Liu [22] applied a Pareto-based multi-objective GA to optimize the sheet metal forming process. In their proposed optimal model, blank-holding force and draw-bead restraining force were optimized as design variables in order to simultaneously minimize the objective functions of fracture, wrinkle, insufficient stretching, and thickness varying. The results showed that this approach is more effective and accurate than the traditional finite element analysis method and the trial-and-error procedure. Non-dominated sorting GA (NSGA-II) was used by Umeonyiagu [23] for multi-objective optimization of the flexural and tensile strength of bamboo-reinforced concrete material. The optimization objectives were the maximization of flexural and tensile strength, as well as the minimization of cost. The research results showed that the Pareto optimal solution would be an effective design guide for engineers for the optimal design of structures using the cost, and flexural and tensile strength of bamboo-reinforced concrete material as design parameters. Yang [24] used NSGA-II to obtain optimum process parameters during stainless steel 316L hot-wire laser welding. During the optimization process, NSGA-II was employed to search for multi-objective Pareto optimal solutions based on ensemble metamodels. The verification tests indicated that the obtained optimal process parameters were effective and reliable for producing the expected welding results. Han [25] conducted

a multi-objective optimization of a corrugated tube with a multi-channel twisted tape (CMCT) to obtain the optimal performance using RSM and NSGA-II. In Li [26], an efficient optimization methodology via the Taguchi method, RSM, and NSGA-II was proposed to optimize a multi-objective problem in the fiber-reinforced composite injection molding process. The results show that RSM can establish efficient predictive models for finding the product quality optimum. Furthermore, the optimum design parameter values determined by NSGA-II were superior to those of the Taguchi method.

The main objective of this study is to carry out an inverse analysis and multi-objective optimization of an ISF process base on the experimental. To that end, a series of experiments are conducted following the BBD for developing RSM and BPNN models with the tool diameter, spindle speed, step depth, and speed rate as the inputs, and the forming angle and thickness reduction as the outputs. Afterward, the effects of the input process parameters on the performance measures are analyzed through the graphs of the main and interaction effect plots. Furthermore, a multi-objective (maximum forming angle and minimum thickness reduction) optimization utilizing the desirability function method and NSGA-II were performed based on the developed RSM models. For the multi-objective optimization of the ISF process, unlike the traditional Taguchi method or RSM, which can only provide a single optimization combination, the Pareto optimal solutions obtained by NSGA-II in this research will provide engineers and designers with better guidance for actual production applications.

## 2. Materials and Methods

### 2.1. Materials and Experiments

The material used in this study is an AA5052-H32 sheet with a thickness of 1 mm. The chemical composition of the AA5052-H32 alloy is given in Table 1. To obtain the flow stress–strain relation, a series of uniaxial tensile tests were performed following the ASTM-8 standard procedure at a constant tensile speed of 3 mm/min with a gauge length of 50 mm. All experiments were processed by a 3D digital image correlation (DIC) system. Through the uniaxial tensile tests, the mechanical properties of AA5052-H32 were obtained, as presented in Table 2.

**Table 1.** Nominal and actual chemical composition of AA5052-H32 alloy (%).

| Composition | Cr | Mg | Si | Cu | Mu | Zn | Fe | Al |
|---|---|---|---|---|---|---|---|---|
| Nominal | 0.18 | 2.23 | 0.14 | 0.01 | 0.05 | 0.001 | 0.31 | Remaining |
| Actual | 0.15 | 2.24 | 0.25 | 0.10 | 0.10 | 0.10 | 0.40 | |

**Table 2.** Mechanical properties of AA5052-H32 alloy.

| Direction | 0° | 45° | 90° |
|---|---|---|---|
| Young's modulus [GPa] | 69.53 | 69.39 | 70.22 |
| Yield stress [MPa] | 165.3 | 154.8 | 156.2 |
| Ultimate tensile strength [MPa] | 223.8 | 215.1 | 218.3 |
| Elongation [%] | 11.2 | 14.4 | 12.1 |
| *R*-value | 0.697 | 0.562 | 0.946 |

The ISF experiments were carried out using a 3-axis computer numerical control (CNC) vertical milling machine (NEXMECCA Inc., Korea) in this research (Figure 2). At the same time, an AA5052-H32 specimen with a size of 130 mm × 130 mm was cut for forming. For lubrication, slide way oil with a kinematic viscosity of 68 was used. Simultaneously, the profile forming tool path used in this study is shown in Figure 3. All the experiments were completed following the experiment design, which is detailed in Section 2.3.

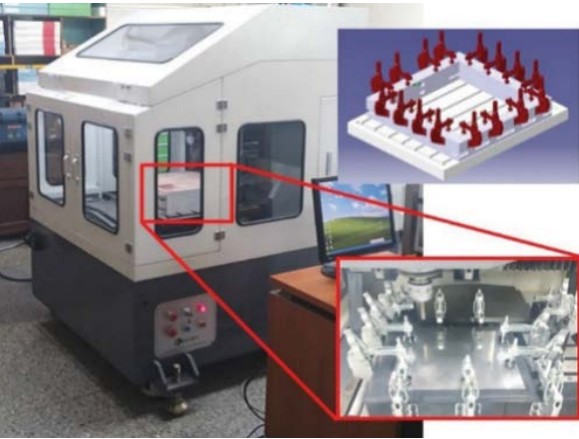

**Figure 2.** Computer numerical control (CNC) machine used in this study.

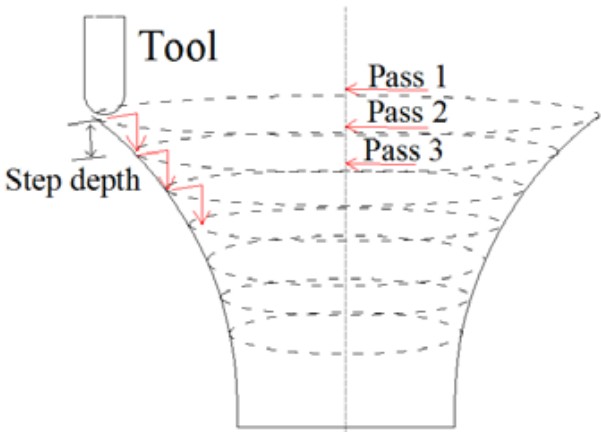

**Figure 3.** Profile tool path.

### 2.2. Formability and Thickness Reduction Measurements

In general, the maximum forming angle is used to evaluate formability in ISF process. In this study, the model of varying wall-angle conical frustums (VWACF) [27–32] was used to obtain the maximum forming angle. As shown in Figure 4, in the VWACF model, as the forming depth increases, the forming angle gradually increases from 40° to 90°. After forming until fracture, the maximum forming angle $\varnothing$ can be obtained using the following formulas:

$$H = L - D + r \tag{1}$$

$$\varnothing = \frac{\pi}{2} - arcsin\left(\frac{H}{r + R}\right) \tag{2}$$

where $D$ is the depth when fracture happens, $R$ is 50 mm, $L$ is 38.3 mm, and $r$ is the tool diameter.

The ISF experiments were performed for the VWACF model according to the experimental conditions. As shown in Figure 4, forming was performed until a fracture occurred in the specimen at the bottom of the model during each experiment. After the forming process, the sheet was cut by the wire cutting to expose the cross-section. As shown in Figure 5a,b, the formed thickness $t$ of all the sheets was measured using a video microscope at a depth of 15 mm from the top plane. The thickness reduction $\Delta t$ can be derived as follows:

$$\Delta t = 1 - t \tag{3}$$

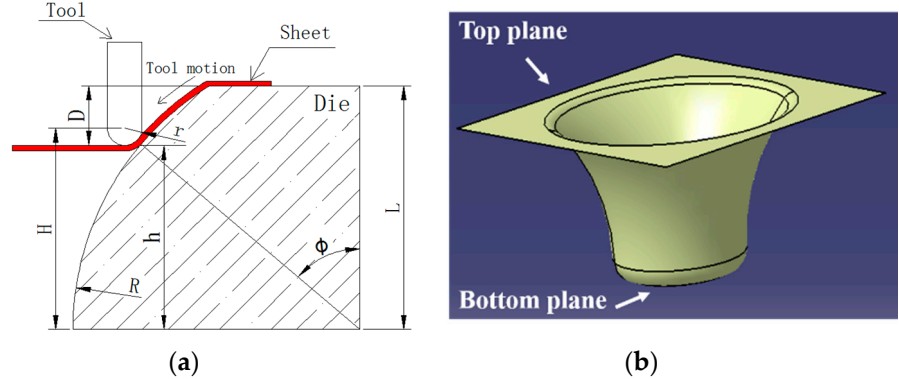

(a)                                          (b)

**Figure 4.** Varying wall-angle conical frustums (VWACF) model: (**a**) 2D schematic diagram and (**b**) 3D schematic diagram for VWACF model.

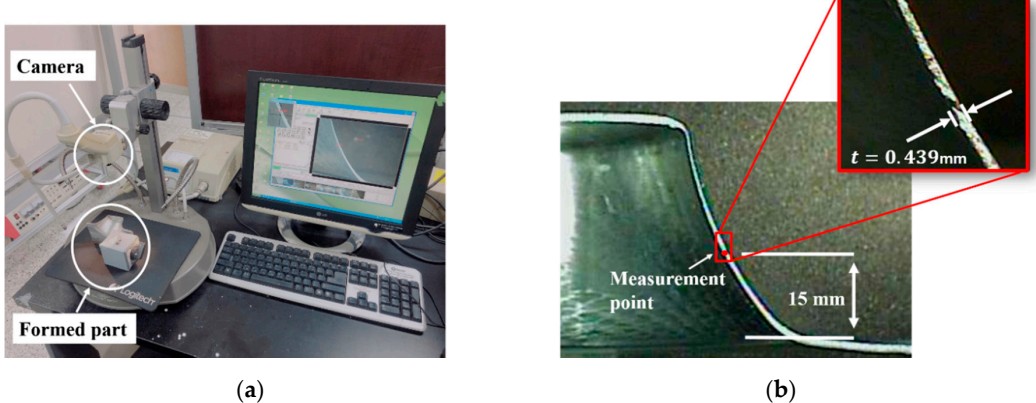

(a)                                          (b)

**Figure 5.** (**a**) Video microscope system and (**b**) measurement of formed thickness for experiment 1.

## 2.3. Experimental Design

BBD is one of the best experiment designs in RSMs and was used to optimize the production process parameters in this research. As mentioned in the introduction, the main forming parameters of the ISF process include tool diameter, step depth in the *z*-direction, sheet thickness, forming temperature, forming speed (feed speed, spindle speed), forming tool path, and others. From previous research reported in literature and from our own experience, it transpires that the tool diameter and step depth in the *z*-direction have the greatest influence on the formability and thickness distribution at room temperature. In addition, the forming time is a long-standing problem in the ISF process, and it is also necessary to optimize the processing speed, while satisfying the quality requirements. Therefore, in this study, the tool diameter, step depth in the *z*-direction, and forming speed were selected as variable parameters. Their selected values and corresponding levels are shown in Table 3. According to BBD, a total of 27 experiments were designed and the results of each experiment are shown in Figure 6, while Table 4 shows the details of the design matrix of the parameters in the actual units employed in the RSM along with the observed responses for forming angle $\varnothing$ and thickness reduction $\Delta t$. Furthermore, the forming force in the *z*-direction of the ISF process for the VWACF shape under different combinations of variables was evaluated in the preliminary experiment. As shown in Figure 7, the maximum vertical force component ($F_z$) for the AA5052-H32 sheet with a thickness of 1 mm is about 1000 N. At the same time, due to the significant influence of the feed speed and step depth, the forming time ranges from 4 to 74 min.

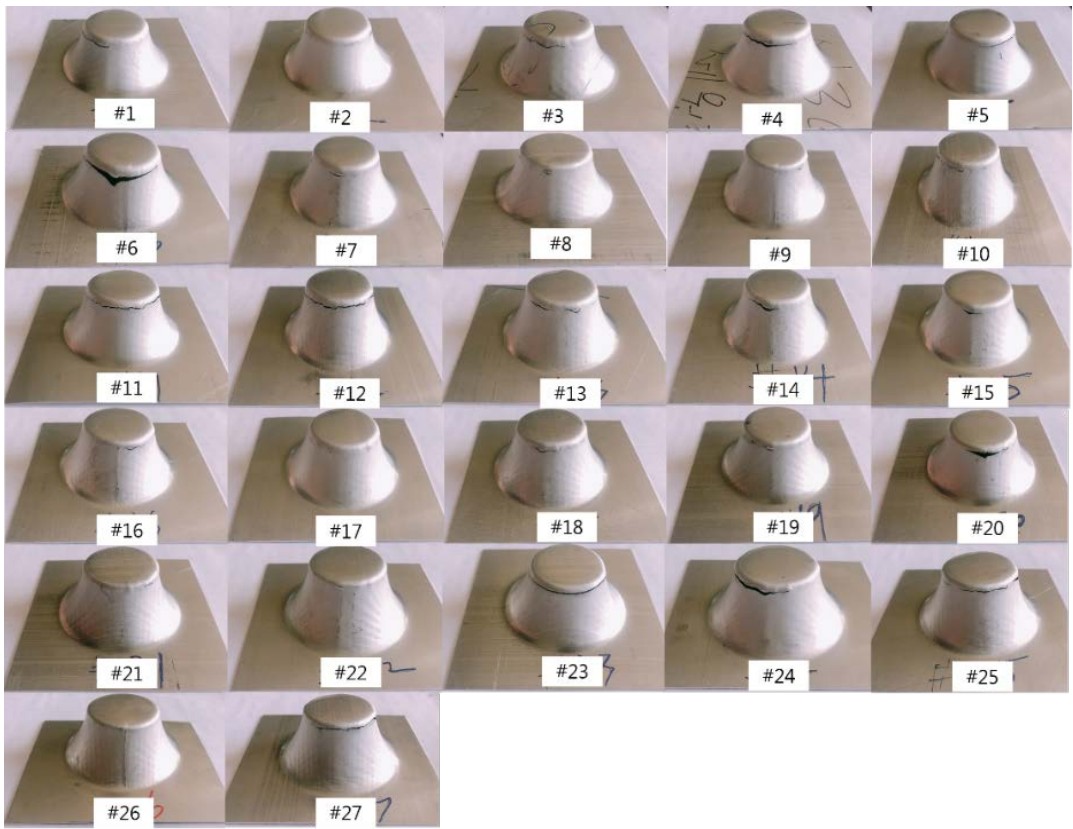

**Figure 6.** Result of 27 experiments for VWACF.

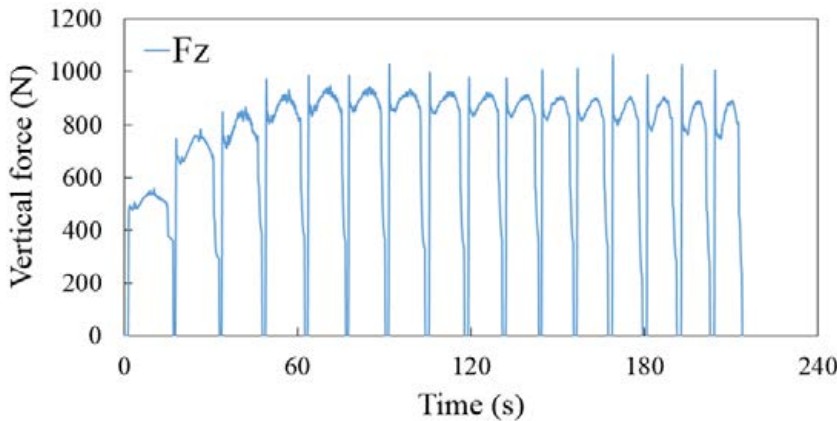

**Figure 7.** Vertical force component ($F_z$) recorded in experiment 18: (tool diameter 6 mm, spindle speed 180 rpm, step depth 0.6 mm, and feed rate 1200 mm/min).

**Table 3.** Level of selected parameters and their levels.

| Parameter | Notable | Unit | Levels | | |
|---|---|---|---|---|---|
| | | | **1** | **2** | **3** |
| | | | **−1** | **0** | **1** |
| Tool diameter | A | mm | 6 | 8 | 10 |
| Spindle speed | B | rpm | 60 | 120 | 180 |
| Step depth | C | mm | 0.2 | 0.4 | 0.6 |
| Feed rate | D | mm/min | 400 | 800 | 1200 |

**Table 4.** Design of experiments and measured responses.

| Exp. No | A | B | C | D | Maximum Forming Angle (°) | Thickness Reduction (mm) |
|---|---|---|---|---|---|---|
| 1 | 0 | −1 | −1 | 0 | 79.704 | 0.561 |
| 2 | 0 | 1 | 0 | −1 | 80.969 | 0.536 |
| 3 | 1 | 0 | 1 | 0 | 79.974 | 0.559 |
| 4 | 0 | 0 | 0 | 0 | 79.541 | 0.569 |
| 5 | 0 | 1 | 0 | 1 | 79.974 | 0.561 |
| 6 | −1 | 0 | 0 | 1 | 79.541 | 0.567 |
| 7 | 0 | −1 | 0 | 1 | 81.169 | 0.542 |
| 8 | 0 | 1 | 1 | 0 | 81.169 | 0.548 |
| 9 | 1 | −1 | 1 | 0 | 80.769 | 0.558 |
| 10 | 1 | 0 | 0 | −1 | 81.169 | 0.551 |
| 11 | −1 | 0 | −1 | 0 | 80.704 | 0.558 |
| 12 | 1 | 0 | −1 | 0 | 79.974 | 0.557 |
| 13 | 0 | 0 | 1 | −1 | 81.169 | 0.541 |
| 14 | 1 | 0 | 0 | 1 | 79.974 | 0.565 |
| 15 | 1 | 1 | 0 | −1 | 81.624 | 0.535 |
| 16 | 0 | 1 | −1 | 0 | 80.704 | 0.548 |
| 17 | 0 | 0 | −1 | 1 | 81.169 | 0.538 |
| 18 | −1 | 1 | 1 | 1 | 80.704 | 0.545 |
| 19 | −1 | 0 | 1 | 0 | 80.704 | 0.547 |
| 20 | −1 | −1 | 0 | 0 | 80.769 | 0.56 |
| 21 | 0 | −1 | 1 | 0 | 80.704 | 0.552 |
| 22 | 0 | −1 | 0 | −1 | 81.169 | 0.54 |
| 23 | 0 | 0 | −1 | −1 | 79.704 | 0.563 |
| 24 | −1 | −1 | −1 | −1 | 79.974 | 0.558 |
| 25 | 1 | 0 | −1 | 1 | 79.974 | 0.55 |
| 26 | −1 | 0 | 0 | 0 | 81.169 | 0.549 |
| 27 | 0 | 0 | 1 | 1 | 79.704 | 0.56 |

## 3. Modeling of Process

### 3.1. RSM Modeling

RSM is a type of statistical technique useful for the modeling of any output of interest as a function of the contributing independent input variables. Following the RSM, an empirical relationship obtained is generally a polynomial, which includes interaction terms:

$$Y = \alpha_0 + \sum_{i=1}^{k} \alpha_i X_i + \sum_{ij=1}^{k} \alpha_{ij} X_i X_j + \sum_{i=1}^{k} \alpha_{ii} X_{ii}^2 + \ldots \tag{4}$$

where the parameters $\alpha$ are the regression coefficients, which are obtained using a least square error minimization technique. A geometrical interpretation of the response function is its corresponding response surface. Using the response surface, the variation in the responses or dependent variables with respect to the independent factors can be presented graphically. The response surface analysis is then carried out utilizing the fitted approximate surface. If the fitted surface is an adequate approximation of the true response function, the analysis of the fitted surface will be approximately equivalent to that of the actual process. In this study, the adequacy of the RSM models was checked through the analysis of residual variances and coefficient of determination ($R^2$).

The coefficients of the RSM regression equation (Equation (3)) were calculated using Design-Expert software. The model equations for the responses (forming angle $\varnothing$ and thickness reduction $t$) to the input parameters $A$, $B$, $C$, and $D$ as listed in Table 2 are as follows:

$$\varnothing = 80.63 - 0.72A - 0.014B - 0.47C + 0.017D - 0.14A^2 + 0.044B^2 - 0.063C^2 - 0.024D^2$$
$$-0.077AB + 0.12AC - 0.035AD + 0.0044BC + 0.018BD - 0.015CD \tag{5}$$

$$t = 0.55 + 0.0075A - 0.001137B + 0.01C - 0.000206D + 0.00445A^2 + 0.00174B^2 - 0.00072C^2$$
$$-0.00185D^2 + 0.00131AB - 0.00262AC - 0.00189AD + 0.00195BC + 0.00171BD - 0.00116CD \tag{6}$$

The ANOVA values used to derive Equation (4) for the forming angle are shown in Table 5. The *F*-value of 32.18 implies the model is significant and there is only a 0.01% chance that an *F*-value this large could occur due to noise. The values in column Prob > *F* less than 0.05 indicate that the model terms are significant, thus, *A*, *C*, and *AA* are significant model terms. In this study, four types of relational expressions dependent on each parameter were constructed—linear, quadratic, 2FI, and cubic. The adequacy measures included, $R^2$, adjusted $R^2$, and predicted $R^2$, are shown in Table 6. All the adequacy measures are in logical agreement and indicate a significant relationship. The ANOVA results for the forming angle model show that the quadratic model captures the best the effect of the four forming parameters (tool diameter, spindle speed, step depth, and feed rate), which, along with the interaction effects of the four parameters, are significant model terms. According to the ANOVA results, all the developed regression models are adequate for quantifying the relationship between process factors and corresponding responses.

**Table 5.** ANOVA for forming angle model.

| Source | Df | Sum of Square | Mean Square | *F*-Value | *P*-Value Prob > *F* | Significant |
|--------|----|--------------|-------------|-----------|----------------------|-------------|
| Model | 14 | 10.0953 | 0.72109 | 32.18 | <0.0001 | Significant |
| *A* | 1 | 5.8935 | 5.89345 | 263 | <0.0001 | Significant |
| *B* | 1 | 0.002 | 0.00201 | 0.09 | 0.769 | |
| *C* | 1 | 2.9176 | 2.91762 | 130.2 | <0.0001 | Significant |
| *D* | 1 | 0.0039 | 0.00389 | 0.17 | 0.684 | |
| *AB* | 1 | 0.0208 | 0.02081 | 0.93 | 0.354 | |
| *AC* | 1 | 0.0745 | 0.07445 | 3.32 | 0.093 | |
| *AD* | 1 | 0.0067 | 0.0067 | 0.3 | 0.594 | |
| *BC* | 1 | 0.0001 | 0.00009 | 0 | 0.949 | |
| *BD* | 1 | 0.0015 | 0.0015 | 0.07 | 0.8 | |
| *CD* | 1 | 0.0012 | 0.00115 | 0.05 | 0.824 | |
| *AA* | 1 | 0.1071 | 0.10713 | 4.78 | 0.049 | Significant |
| *BB* | 1 | 0.0091 | 0.00906 | 0.4 | 0.537 | |
| *CC* | 1 | 0.0165 | 0.01645 | 0.73 | 0.408 | |
| *DD* | 1 | 0.0024 | 0.00242 | 0.11 | 0.748 | |
| Residual | 12 | 0.2689 | 0.02241 | | | |
| Cor Total | 26 | 10.3642 | | | | |

**Table 6.** Model summary for forming angle response.

| Source | Std. Dev | *R* Squared | Adjusted *R* Squared | Predicted *R* Squared | Press | Suggested |
|--------|----------|-------------|----------------------|-----------------------|-------|-----------|
| Linear | 0.164015 | 0.942897 | 0.932515 | 0.91355 | 0.895982 | |
| **Quadratic** | **0.149696** | **0.974054** | **0.943784** | **0.881749** | **1.225576** | **Suggested** |
| 2FI | 0.167485 | 0.956695 | 0.92963 | 0.881063 | 1.232687 | |
| Cubic | 0.173 | 0.997112 | 0.924919 | | | Aliased |

Table 7 shows the ANOVA results for the thickness reduction model, where the *F*-value of 17.48 implies that the model is significant. For the thickness reduction, *A*, *C*, and *AA* are significant model terms. Adequacy measures, $R^2$, adjusted $R^2$, and predicted $R^2$ are shown in Table 8, which shows the quadratic model is also the best for the thickness reduction.

**Table 7.** ANOVA for thickness reduction model.

| Source | Df | Sum of Square | Mean Square | *F*-Value | *P*-Value Prob > *F* | Significant |
|---|---|---|---|---|---|---|
| Model | 14 | 0.00238 | 0.00017 | 17.47566 | <0.0001 | Significant |
| *A* | 1 | 0.00065 | 0.00065 | 66.90042 | <0.0001 | Significant |
| *B* | 1 | 0.000014 | 0.000014 | 1.463945 | 0.2496 | |
| *C* | 1 | 0.00134 | 0.00134 | 137.9412 | <0.0001 | Significant |
| *D* | 1 | 0.0000006 | 0.0000006 | 0.056772 | 0.8157 | |
| *AB* | 1 | 0.000006 | 0.000006 | 0.615023 | 0.4481 | |
| *AC* | 1 | 0.000037 | 0.000037 | 3.829763 | 0.0740 | |
| *AD* | 1 | 0.0000195 | 0.0000195 | 2.007373 | 0.1820 | |
| *BC* | 1 | 0.000018 | 0.000018 | 1.870631 | 0.1965 | |
| *BD* | 1 | 0.000014 | 0.000014 | 1.399562 | 0.2597 | |
| *CD* | 1 | 0.0000066 | 0.0000066 | 0.674061 | 0.4276 | |
| *AA* | 1 | 0.0001 | 0.0001 | 10.41962 | 0.0072 | Significant |
| *BB* | 1 | 0.000014 | 0.000014 | 1.429612 | 0.2549 | |
| *CC* | 1 | 0.000002 | 0.000002 | 0.219143 | 0.6481 | |
| *DD* | 1 | 0.000014 | 0.000014 | 1.48132 | 0.2470 | |
| Residual | 12 | 0.000117 | 0.0000097 | | | |
| Cor Total | 26 | 0.002495 | | | | |

**Table 8.** Model summary for thickness reduction response.

| Source | Std. Dev | *R* Squared | Adjusted *R* Squared | Predicted *R* Squared | Press | Suggested |
|---|---|---|---|---|---|---|
| Linear | 0.004302 | 0.836777 | 0.8071 | 0.753348 | 0.000615 | |
| **Quadratic** | **0.003118** | **0.953245** | **0.89869** | **0.752982** | **0.000616** | **Suggested** |
| 2FI | 0.004214 | 0.886135 | 0.81497 | 0.688526 | 0.000777 | |
| Cubic | 0.00125 | 0.999374 | 0.983716 | | | Aliased |

*3.2. Parametric Influence*

3.2.1. Analysis of Forming Angle Effect on Process Parameters

The relationships between the forming angle and four forming parameters are shown in Figure 8a–c. Normal probability plots of residuals for forming angle are shown in Figure 8a. It was observed that the residuals were normally distributed as most of them were clustered around the straight reference line. It was observed that the regression model fitted the observed values fairly well. Figure 8b shows the perturbation plot of the four factors influencing the forming angle response. The perturbation plot is an important diagrammatic representation to compare the effects of all factors at a particular point in the design space. The response is plotted by changing only one factor over its range while holding other factors constant. From Figure 8b, as the results show, the input parameters tool diameter and step depth have a significant effect on the forming angle; the tool diameter has the most effect on the forming angle. The plot of the experimental responses versus predicted responses shown in Figure 8c demonstrates that there is a very good correlation between the observed value and the values predicted by the model.

Figure 9 shows the three-dimensional contour plot of the effect of each parameter on the forming angle. Each plot shows the effect of two process variables within their experimental study ranges with the other variable fixed at the central point values. Combined with Figure 8b, the experimental results prove that a smaller tool radius enables a higher forming angle than can be achieved by a larger one. In the case of a small tool radius, there is a highly concentrated zone of deformation that causes high strain and leads to better formability. It is also found that a decrease in step depth cause a higher forming angle. This is because a larger step depth will generate a pulling effect caused by a large tensile force along the wall of the formed ISF part, which will compromise the stabilization effect from the bending in the contact area. The impact of the tool on the sheet resulting from the high speed

movement of the tool will make the pulling effect even more pronounced. However, the forming time will increase as the step depth decreases; thus, a smaller step depth should be used within a reasonable range in the tool path control and optimization.

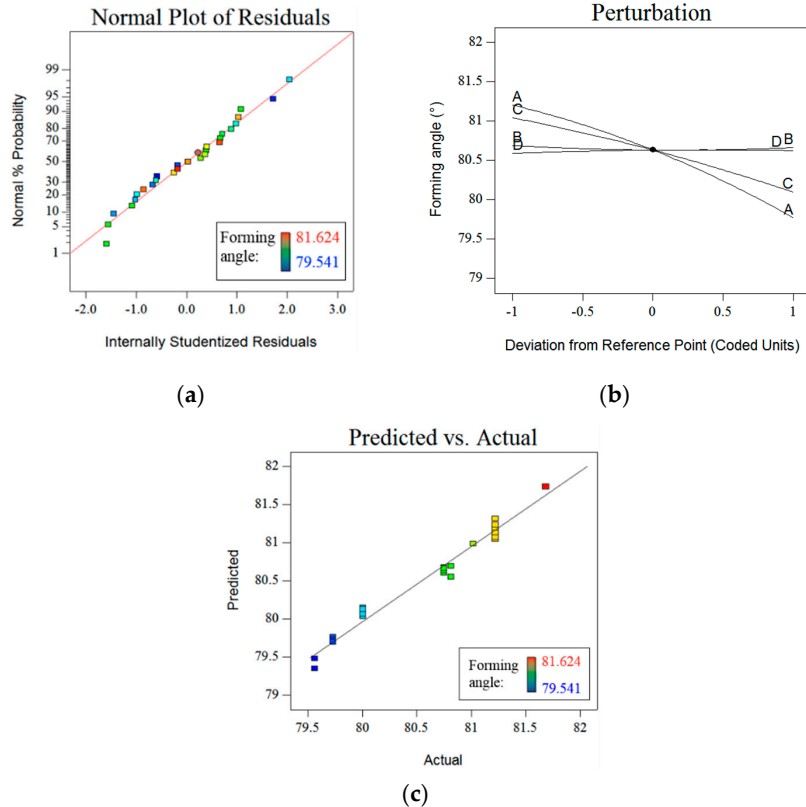

(a)  (b)

(c)

**Figure 8.** (**a**) Normal plot of forming angle residuals with central reference point 0.00, (**b**) perturbation plot of forming angle with central reference point 0.00, and (**c**) predicted vs. actual values of forming angle.

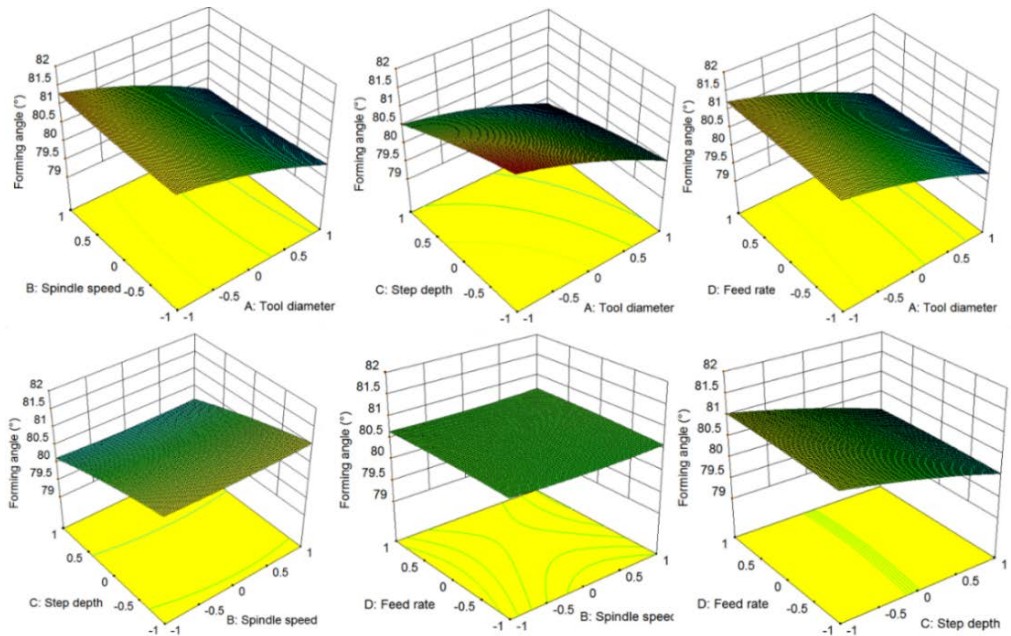

**Figure 9.** Three-dimensional contour plots of effect of each parameter on forming angle.

In addition, the feed rate and spindle speed are found to have almost no significant effect on the forming angle in this case. Usually, the forming angle increases along with the forming speed due to heating caused by the relative motion and friction between the tool and the blank. However, in this work, the spindle speed and feed rate are not high enough to cause significant heating effects to improve the forming angle.

### 3.2.2. Thickness Reduction

The relationships between the thickness reduction and forming parameters are shown in Figure 10a–c. Normal probability plots of residuals for the thickness reduction demonstrate that they were normally distributed, as shown in Figure 10a. From Figure 10b, the input parameters tool diameter and step depth have a significant effect on the thickness reduction, while the step depth has the largest effect on the thickness reduction. The plot of experimental versus predicted responses shown in Figure 10c demonstrated that there is also a very good correlation between the observed and the values predicted by the model.

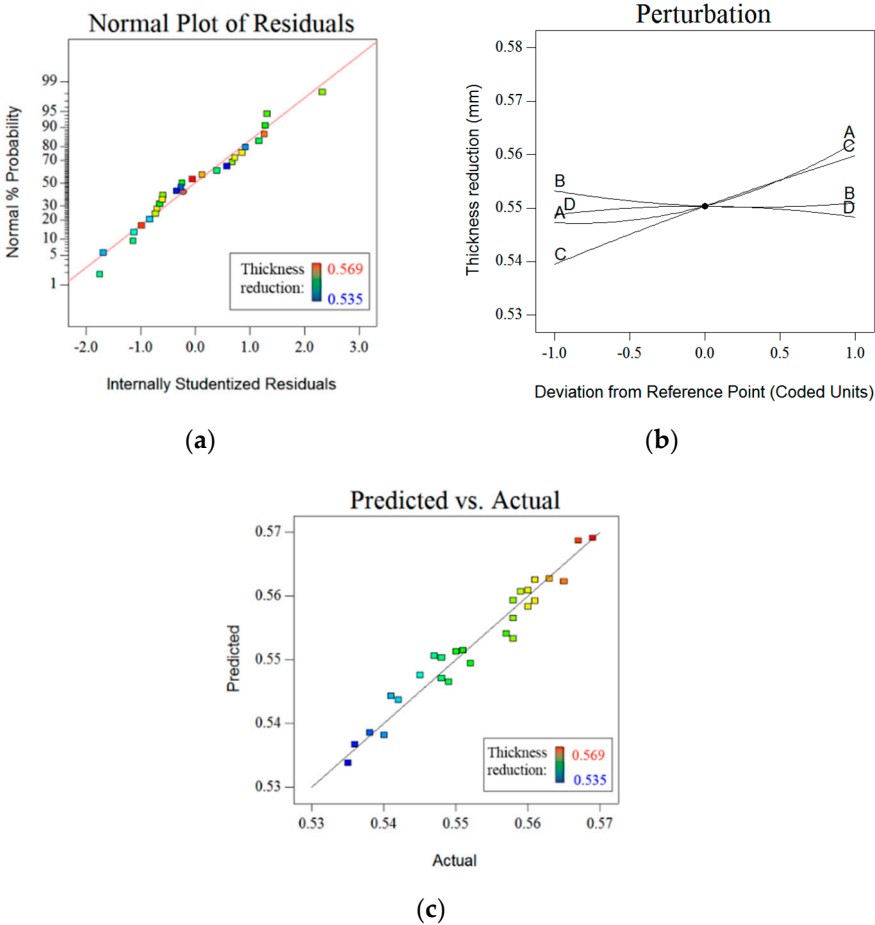

**Figure 10.** (**a**) Normal plot of thickness reduction residuals with central reference point 0.00, (**b**) perturbation plot of thickness reduction with central reference point 0.00, and (**c**) predicted vs. actual values of thickness reduction.

Figure 11 shows the three-dimensional contour plots of the effect of each parameter on the thickness reduction. In addition, combined with Figure 10b, an increase in tool diameter increases the thickness reduction. During the ISF process, the surface between the sheet and the tool is affected by friction due to the sliding and rolling motion of the tool. The friction increases with the tool diameter increase because the wider area is subjected to friction. This causes an increase in temperature on

the interface of the tool and the sheet that enhances the ductility of the material. In such conditions, tension increases in the material during forming and causes higher thickness reduction. It is also found from the figure that an increase in the step depth causes higher thickness reduction. During the ISF process, for higher value of step depth, the tension due to bending of the sheet material also increases and causes a sudden reduction in thickness that restricts formability, which leads to further thickness reduction.

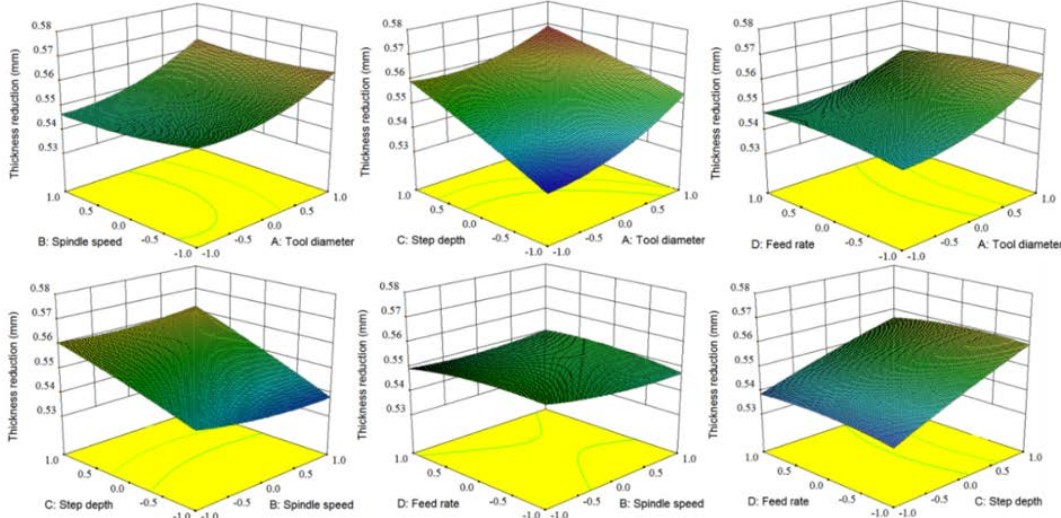

**Figure 11.** Three-dimensional contour plots effect of each parameter on thickness reduction.

At the same time, it can be noticed that the feed rate has almost no effect on the thickness reduction. That is because the range of feed rate changes is too small to cause the effects to change the thickness reduction.

### 3.3. Multi-Objective Optimization Using RSM

Table 9 shows the results of the response optimization solution including single-response and composite desirability obtained using Design-Expert software. From Table 7, the optimal result is achieved when the tool diameter is 6 mm, spindle speed is 120 rpm, step depth is 0.2 mm, and feed rate is 800 min/mm, respectively, with which the maximum forming angle is 81.711° and the minimum thickness reduction is 0.534 mm, respectively.

**Table 9.** Result of optimization.

| Tool Diameter (mm) | Spindle Speed (RPM) | Step Depth (mm) | Feed Rate (mm/min) | Maximum Forming Angle (°) | Minimum Thickness Reduction (mm) |
|---|---|---|---|---|---|
| 6 | 120 | 0.2 | 800 | 81.711 | 0.534 |

### 3.4. BPNN Modeling

A typical BPNN comprises an input layer, hidden layers, and an output layer (Figure 12). Depending on the nonlinearity and complexity of the model, it may have several hidden layers of neurons. The neurons in each layer add up the values delivered from the previous layer, process them, and pass them to the next layer of neurons. Finally, the final value obtained at the output layer is compared with the sample data obtained from the test, the error is analyzed, and the learning process is conducted by adjusting the weights of each neuron by the backpropagation method.

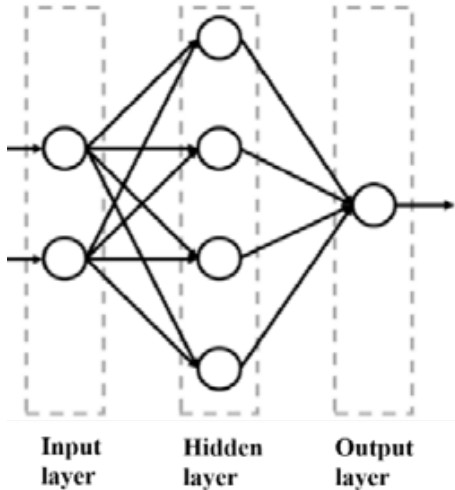

**Figure 12.** Schematic of back propagation neural network.

The weighted sum from the input layer to the hidden layer:

$$net_k = b + w^T x = b_k + \sum w_{kj} x_j \tag{7}$$

where $k$ and $j$ are the number of hidden layers and input variables, respectively, $b$ is a bias vector, $w$ is a weight between each neuron, and $x$ is an input vector. The value derived in the above process is applied to a nonlinearity activation function, thereby predicting the forming angle and thickness reduction observed in the VWACF experiment. In this study, the sigmoid function and the identity function are used as the activation function for the input layer, and the forming angle $\overline{\varphi}_{angle}$ angle can be derived as follows.

$$\text{Sigmoid}: \ f(net_k) = \left\{ \frac{2}{[1 + exp(-2 * net_k)]} \right\} - 1 \tag{8}$$

$$\text{Identity}: \ \overline{\varphi}_{angle} = f(net_k) = max(0, net_k) = \varphi_{predicted} \tag{9}$$

Furthermore, the mean square error (MSE) was defined to compare the difference between the predicted angle and the actual angle as follows:

$$E = 1/Q \left\{ \sum \left[ \overline{\varphi}_{actual}(m) - \varphi_{predicted}(m) \right]^2 \right\} \tag{10}$$

where $Q$, $\overline{\varphi}_{actual}$, $\varphi_{predicted}$ are the number of data, forming angle obtained in the experiment, and the forming angle predicted by the neural network, respectively. In this research, the Levenberg–Marquardt approach [33], which is one of the backpropagation techniques, was used to train the BPNN. The Levenberg–Marquardt technique is given by the following formula:

$$w_{i+1} = w_i - [J^T J + \mu_i diag(J^T J)]^{-1} J^T n \tag{11}$$

where $w$ is the weight of each neuron, $i$ is the number of repetitions, $J$ is the Jacobian matrix, $\mu$ is the damping factor, and $n$ is the residual between the actual forming angle and the forming angle predicted by the neural network. In this case, the Jacobian matrix is defined using the backward difference method as follows:

$$J = \frac{\partial r(w)}{\partial w} = \frac{r_1 - r_{i-1}}{\Delta w} \tag{12}$$

In Equation (10), $\mu_i diag(J^T J)$ is a diagonal matrix extracted from the Hessian matrix, i.e., the matrix of eigenvalues of the Hessian matrix, or the curvatures. The Levenberg–Marquardt technique was

proposed to solve the step size problem in the Levenberg method, which was an improvement on the Gauss–Newton method. In summary, the neurons of the BP network learn to minimize errors through the above iterative process.

In this study, the tool diameter, spindle speed, step depth, and feed rate were applied as input variables, and the forming angle and thickness reduction were used as output variables. After many trials, a hidden layer with ten neurons was found to be the most suitable. Figure 13 shows the structure of the BPNN used in this research.

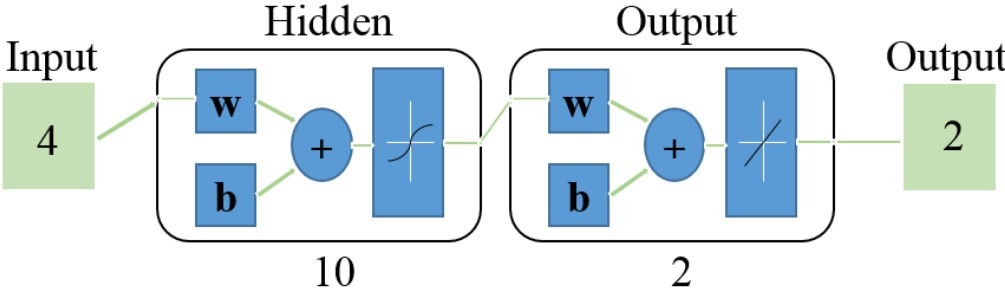

**Figure 13.** Architecture of the back propagation neural network (BPNN) for forming angle and forming depth prediction.

In this model, we randomly selected 19 out of the 27 datasets for training (70% of total data) and four validation set and testing (15% of data), respectively. Figure 14a,b shows the correlation between the experimentally obtained forming angle and thickness and their BPNN-predicted values. As shown in Figure 14a,b, the correlation coefficients for the forming angle and the thickness reduction are 0.99769 and 0.97546, respectively, which are both close to the target of 1.

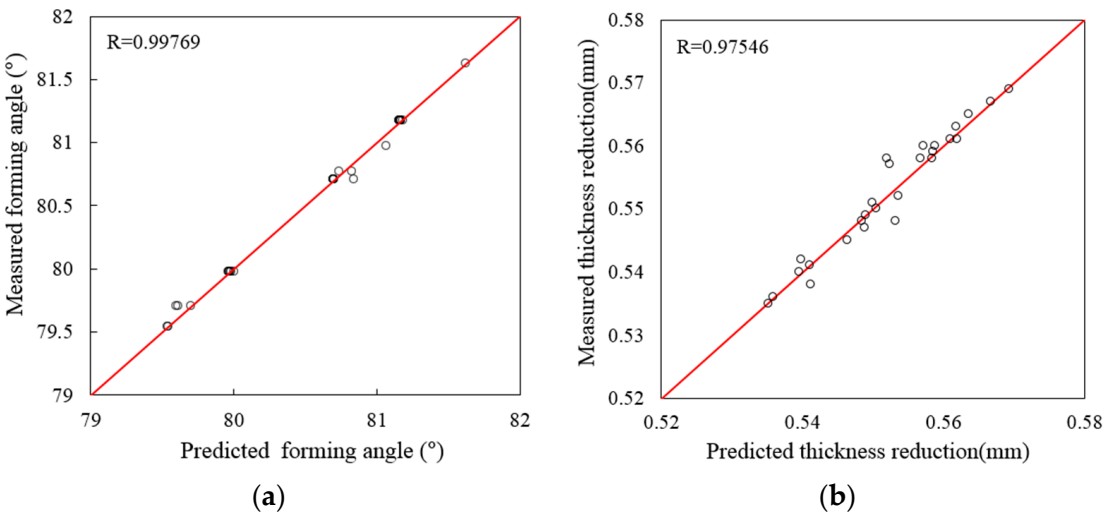

(a)         (b)

**Figure 14.** Correlation among training set data: (**a**) forming angle and (**b**) thickness reduction.

The forming angle and thickness reduction predicted using the BPNN were calculated and summarized in Table 10. When comparing the predicted forming angle and thickness with their actual values, the average error was about 0.03% and 0.26%, respectively, which confirmed that the forming angle and thickness reduction can be accurately estimated using the proposed model.

**Table 10.** Results of BPNN.

| Exp. No | Actual Forming Angle (°) | Predicted Forming Angle (°) | Actual Thickness Reduction (mm) | Predicted Thickness Reduction (mm) |
|---|---|---|---|---|
| 1 | 79.704 | 79.614 | 0.561 | 0.561 |
| 2 | 80.969 | 81.072 | 0.536 | 0.536 |
| 3 | 79.974 | 79.987 | 0.559 | 0.559 |
| 4 | 79.541 | 79.541 | 0.569 | 0.569 |
| 5 | 79.974 | 79.976 | 0.561 | 0.562 |
| 6 | 79.541 | 79.544 | 0.567 | 0.567 |
| 7 | 81.169 | 81.164 | 0.542 | 0.54 |
| 8 | 81.169 | 81.184 | 0.548 | 0.553 |
| 9 | 80.769 | 80.740 | 0.558 | 0.557 |
| 10 | 81.169 | 81.169 | 0.551 | 0.55 |
| 11 | 80.704 | 80.698 | 0.558 | 0.552 |
| 12 | 79.974 | 79.972 | 0.557 | 0.552 |
| 13 | 81.169 | 81.168 | 0.541 | 0.541 |
| 14 | 79.974 | 79.974 | 0.565 | 0.564 |
| 15 | 81.624 | 81.628 | 0.535 | 0.535 |
| 16 | 80.704 | 80.699 | 0.548 | 0.548 |
| 17 | 81.169 | 81.171 | 0.538 | 0.541 |
| 18 | 80.704 | 80.705 | 0.545 | 0.546 |
| 19 | 80.704 | 80.845 | 0.547 | 0.549 |
| 20 | 80.769 | 80.826 | 0.56 | 0.559 |
| 21 | 80.704 | 80.695 | 0.552 | 0.554 |
| 22 | 81.169 | 81.161 | 0.54 | 0.54 |
| 23 | 79.704 | 79.705 | 0.563 | 0.562 |
| 24 | 79.974 | 79.962 | 0.558 | 0.558 |
| 25 | 79.974 | 80.004 | 0.55 | 0.551 |
| 26 | 81.169 | 81.159 | 0.549 | 0.549 |
| 27 | 79.704 | 79.602 | 0.56 | 0.557 |

## 4. Multi-Objective Optimization

The objective of this research was to achieve the minimum thickness reduction while maximizing the forming angle in the ISF process for the AA5052 alloy. These multi-objectives can only be achieved through multi-objective optimization. Because changing one objective necessarily affects another objective, optimization does not have a single solution, but rather a series of solutions on a Pareto front called the non-dominated solutions [34]. The Pareto front solutions are efficient because moving away from the Pareto front in order to improve one objective would lead to making one or more objectives worse off. The Pareto front provides a convenient choice for practical applications, which could be used to select an optimal solution depending on the requirement of the part to be formed in the ISF process.

### 4.1. GA for Optimal Pareto Front

GAs are computational models developed to mirror the evolution of the natural world and are global optimizers [35,36]. In the GA model developed for this study, the process input parameters (tool diameter, step depth, spindle speed, and speed rate) constitute the genes (chromosomes). Various combinations of the genes (chromosomes) constitute a population. The GA procedure involves first generating an initial population and then using the fitness function to assess the fitness of the population members. After the assessment, the most suitable population members are selected for transfer to the next generation. Usually, about half of the initial population is chosen for this purpose. The selected population undergoes cross breeding and mutation in the second generation, and during the process, the population size is culled back to the initial size. Similarly, half of the population of the second generation will be chosen to propagate to the next third generation based on their fitness. The population regeneration process occurs again in the third generation through cross breeding and mutation. The process continues until the optimum or close to the optimum solution is achieved.

A variant of the GA procedure for multi-objective optimization called the non-dominated sorting genetic algorithm II (NSGA-II) is used in this research. As shown in Figure 15, the evaluation of the

fitness of the population members in NSGA-II is done using a fitness function, which relates the input parameters to the outputs, given by Equations (4) and (5) in this research. The RSM-NSGA-II algorithm multi-objective optimization system is shown in Figure 16.

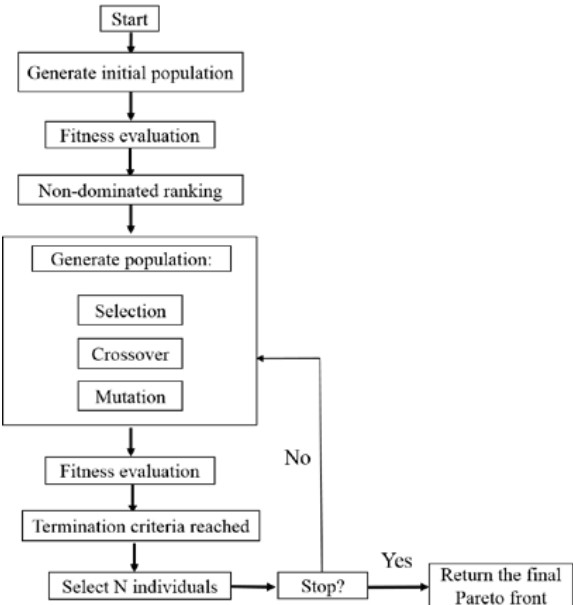

**Figure 15.** Process of the non-dominated sorting genetic algorithm II (NSGA-II) algorithm.

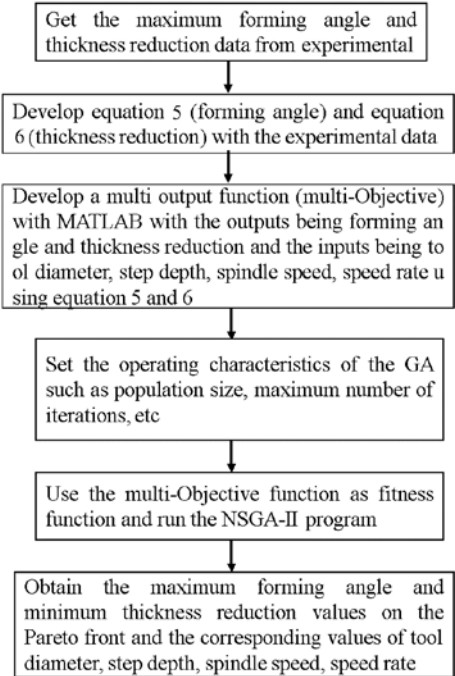

**Figure 16.** Response surface method (RSM)-NSGA-II algorithm for multi-objective optimization.

### 4.2. Optimization Procedure Based on NSGA-II Algorithm

The objectives of the NSGA-II optimization algorithm in this work were the maximization of the forming angle and minimization of the thickness reduction in the ISF process. The upper (1) and lower (−1) levels of the experimental input parameters tool diameter, spindle speed, step depth, and speed rate were used as the limits of the GA algorithm. The population size used by the GA was 50, while the crossover and mutation rates were set at 0.8 and 0.01, respectively. The maximum

number of generations was set to 500 in order to ensure that the algorithm runs to completion. The optimized Pareto front obtained after 102 iterations of the GA is shown in Figure 17, while detailed results of the 18 solution sets are shown in Table 11. Each point from Figure 17 represents a specific optimal solution, and the corresponding input parameters in Table 11 can be selected according to the requirements of the part geometry with a desired forming angle and thickness reduction. For example, the forming condition which was maximum forming angle 81.73° and thickness reduction 0.535 mm is corresponding to solution no. 1 in Table 11, tool diameter is 6 mm, spindle speed is 159 rpm, step depth is 0.2 mm, and feed rate is 1169 mm/min. It can be seen that the difference between the maximum and minimum optimal value of the forming angle was 2.39, while the difference between the maximum and minimum optimal value of thickness reduction was 0.04, respectively.

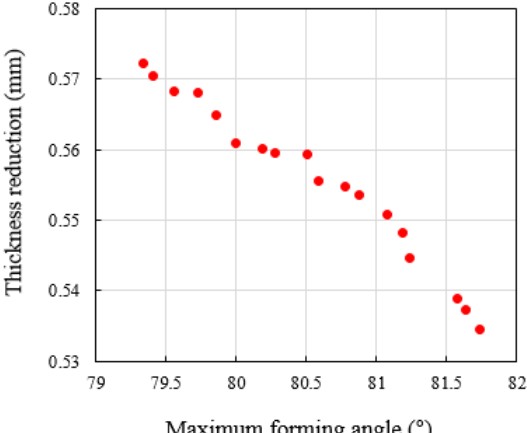

**Figure 17.** Pareto front of optimal process outputs.

**Table 11.** Optimal parameter combination by Pareto front.

| Solution No. | *A* (mm) | *B* (rpm) | *C* (mm) | *D* (mm/min) | Forming Angle (°) | Thickness Reduction (mm) |
|---|---|---|---|---|---|---|
| 1 | 6 | 159 | 0.2 | 1169 | 81.733 | 0.535 |
| 2 | 6 | 86 | 0.6 | 715 | 80.510 | 0.559 |
| 3 | 10 | 176 | 0.6 | 775 | 79.339 | 0.572 |
| 4 | 6 | 165 | 0.5 | 1099 | 81.074 | 0.551 |
| 5 | 6 | 159 | 0.4 | 1153 | 81.184 | 0.548 |
| 6 | 8 | 161 | 0.6 | 873 | 80.277 | 0.560 |
| 7 | 7 | 150 | 0.6 | 1160 | 80.590 | 0.556 |
| 8 | 9 | 93 | 0.5 | 624 | 80.002 | 0.561 |
| 9 | 8 | 162 | 0.6 | 825 | 80.184 | 0.560 |
| 10 | 6 | 147 | 0.2 | 1165 | 81.634 | 0.537 |
| 11 | 7 | 174 | 0.5 | 1153 | 80.772 | 0.555 |
| 12 | 10 | 63 | 0.5 | 529 | 79.729 | 0.568 |
| 13 | 6 | 89 | 0.5 | 915 | 80.876 | 0.554 |
| 14 | 10 | 73 | 0.4 | 630 | 79.854 | 0.565 |
| 15 | 7 | 169 | 0.4 | 1150 | 81.236 | 0.545 |
| 16 | 6 | 159 | 0.3 | 1161 | 81.578 | 0.539 |
| 17 | 10 | 163 | 0.6 | 746 | 79.407 | 0.570 |
| 18 | 9 | 163 | 0.6 | 585 | 79.554 | 0.568 |

### 4.3. Comparison between NSGA-II and RSM

As shown in Table 12, the optimal result from RSM is similar to the first row of the Pareto results shown in Table 11. However, compared to the optimization using the NSGA-II algorithm, the optimization using the RSM method does not produce a Pareto front. It is important to mention here that the Pareto optimal solutions can provide the maximum forming angle for a specified value of deformed sheet thickness reduction of a metal part to be formed in ISF. Conversely, because of its limitations, the optimal values obtained through RSM can mislead designers and engineers.

**Table 12.** Comparison between the optimal results using the RSM and the genetic algorithm (GA).

| Method | Tool Diameter | Spindle Speed | Step Depth | Feed Fate | Maximum Forming Angle (°) | Minimum Thickness Reduction (mm) |
|--------|--------------|---------------|-----------|-----------|---------------------------|----------------------------------|
| RSM | 6 | 120 | 0.2 | 800 | 81.711 | 0.534 |
| GA | 6 | 159 | 0.2 | 1169 | 81.733 | 0.535 |

## 5. Conclusions

In this research, modeling the ISF process included for forming parameters (tool diameter, spindle speed, step depth, and feed rate) using RSM and an ANN algorithm were conducted. Moreover, this research has shown the applicability of GA to optimization of the forming parameters. The following conclusions can be drawn:

- The relationships between the output (forming angle and thickness reduction) and input parameters were established by RSM. According to the results, it was found that second-order polynomial regression models offer a good fit for both the forming angle and the thickness reduction.
- The effect of each parameter on the forming result is analyzed by RSM. Among them, the tool diameter and step depth of the tool have a significant impact on the forming angle and thickness reduction. Increasing the tool diameter and step depth lead the lower formability and pronounced thickness reduction. However, the range of feed rate considered in this research was too small to cause a significant change in the forming results.
- A BPNN was effectively used to model and predict the relationship between the parameters and the response in the ISF process.
- Both RSM and NSGA-II can effectively be used for multi-objective optimization of the ISF forming process. However, an optimal Pareto front solution from the NSGA-II can provide multiple choices, which means it can offer a rational design guide for practical applications of the ISF process.

## 6. Future Works

FEM is employed in the manufacturing industry to predict the behavior of the formed sheet metal component. However, there are many issues associated with FEA when it comes to simulating the ISF process, in which problems such as the accuracy of the FEM results and the long analysis time require more research. Simultaneously, tool path generation is a key topic in the ISF process. It is essential to develop dedicated tool paths to improve the efficiency and accuracy of this process.

**Author Contributions:** X.X. and Y.-S.K. conceived and designed the experiments; S.Y. performed the experiments; M.-P.H. and J.-J.K. analyzed the data; X.X. wrote the paper. All authors have read and agreed to the published version of the manuscript.

**Funding:** This work was supported by the Ministry of Education of the Republic of Korea and the National Research Foundation of Korea (NRF-2019R1A2C1011224) and the Ministry of Trade, Industry and Energy the Global Professional Technology Development Project (20000231).

**Conflicts of Interest:** The authors declare no conflict of interest.

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
