# Peer review of "RSM and BPNN Modeling in Incremental Sheet Forming Process for AA5052 Sheet: Multi-Objective Optimization Using Genetic Algorithm"

_metals, doi:10.3390/met10081003_

Round 1
Reviewer 1 Report
This paper presents a study about the application of RSM and BPNN for modelling the incremental sheet forming process for AA5052 sheet. An experimental campaign designed for four process parameters was carried out. The results obtained in this experimental campaign are used for modelling the variations in the forming angle and thickness reduction of AA52052 sheet. The multi-objective optimization using RSM-based genetic algorithm was then applied to obtain the Pareto optimal solutions for maximum forming angle and minimum thickness reduction. The ISF is an emerging process to manufacture sheet metal and the topic of this paper is of interest to the engineering and researcher community. However, this study lacks originality since many aspects adopted in this study like as the use of RSM or BPNN for modelling ISF or the multi-objective optimization using the GA is very common and can be found in the existing literature. The major contribution of this work compared with the existing published works was not clearly highlighted.
In this regard, the reviewer recommends that the paper is not in a fit state of the original paper for publication in this Journal. This paper may be published as a technical note with the following comments taken into account.
- The major contribution of the study should be highlighted clearly concerning the existing literature.
- The authors have done a state of the art about the topic with some existing published papers. In the Reviewer’s opinion, the references are scarce, since there are many more articles that have worked on the same topic.
- The form of equations (4) and (5) is not easy to interpret. It’s recommendable using the mathematical form.
- English, in general, should be improved.
Author Response
Dear Reviewer:
Thank you for your letter and the reviewer’s comments concerning our manuscript entitled “RSM and BPNN modeling in incremental sheet forming process for AA5052 sheet: Multi-objective optimization using genetic algorithm” (Manuscript ID: metals-838932). Those comments are all valuable and helpful for revising and improving our paper, as well as an important and significant guidance for our research. We have studied your comments carefully and have made corrections, which we hope will meet with your approval.
Please see the attachment for the cover letter.
Once again, thank you very much for your comments and suggestions.

Reviewer 2 Report
The paper presents a commendable work in the field of incremental forming process. The paper is well written, but the real question is the paper mission. The question is why the paper has been developed instead of how it was written.
Before deciding if the paper could be published, there are some issues which have to be addressed by the authors:
- numerical simulation is today able to estimate all the proces parameters (mainly maximum wall angle and thickness reduction);
- moreover, one can find in the literature comprehensive experimental databases with regards of the process parameters. There are also analytical formulas which can accurately predict the maximum wall angle. Even for manufacturing parts showing highly variable wall angle, the formulas (established for constant angle) can be applied locally;
- the processing trajectories (toolpaths) are not considered in the experiments, while many papers form the literature have shown that their influence upon the process parameters (maximum wall angle and thickness reduction) could be significant;
- the method of measuring thickness reduction used in the paper is quite inaccurate (lines 146-147 and figure 4). Usually these measurements are performed using optical measuring devices.
Author Response

(The authors gave the same response as above.)

Reviewer 3 Report
The paper is more interesting from optimization standpoint. Regarding ISF process it does not add much to the existing knowledge on the process.
Page 5: how to ensure that wire cutting does not affect the measured thickness?
line 343: angle not “angel”
Too bad forming force was not subject of optimisation as being a critical factor for machine employed.
Processing time should also enter in the accounting once it is an impairing drawback for incremental sheet forming adoption worldwide. Please report forming times for a number of variants tests, namely the optimized ones. also for the forming forces.
Specify the lubricant used.
Just one material was studied. If a steel material was also used, it would be possible to double-check the method and infer some very nice differences between material responses
Spindle speed optimisation is a misleading parameter to be studied, since several machines (specially the purpose-built ones) have free-rotation, driven only by friction where necessary
Author Response

(The authors gave the same response as above.)

Round 2
Reviewer 1 Report
The authors have addressed all comments. I do not have any further comments.
Author Response
Dear Reviewer:
Thank you for your letter and the reviewer’s comments concerning our manuscript entitled “RSM and BPNN modeling in incremental sheet forming process for AA5052 sheet: Multi-objective optimization using genetic algorithm” (Manuscript ID: metals-838932). we are sure that those comments are all valuable and helpful for improving our research in the future.
Once again, thank you very much for your comments and suggestions.
Reviewer 2 Report
In the first review, the main issues pointed by the reviewer were:
- Numerical simulation is today able to estimate all the process parameters
- One can find in the literature comprehensive experimental databases with regards of the process parameters and also analytical formulas which can accurately predict the maximum wall angle
- The processing trajectories (toolpaths) are not considered in the experiments, while many papers form the literature have shown that their influence upon the process parameters (maximum wall angle and thickness reduction) could be significant
- The method of measuring thickness reduction used in the paper is quite inaccurate, compared with the state of the art (and good practices in the field)
The authors responded to each issue using personal assumptions not backed-up by solid literature references:
- Numerical simulation was considered by authors a time-consuming process compared to experiments (30 hrs compared to 30-40 mins). Time to set-up experiments and costs are not considered by the authors. One literature reference is included.
- The use of varying wall-angle conical frustums (VWACF) was considered by authors as a solution to overcome the issue “there is no analytical formula that can accurately predict the maximum forming angle”, although the literature suggests otherwise. Of course, this is still a debatable issue, but the authors should provide at least some literature references backing-up their point of view
- The authors provided only their point of view regarding why the do not consider toolpaths in their research, but the main question remains about what toolpaths they really used for the experiments. It is not presented in the paper.
- The authors answered to this issue by the following affirmations:
“the forming shape has varying wall-angle conical frustums, which is not a particularly complicated shape, so the thickness distribution is relatively uniform” – measured by what? Based upon the above-mentioned time consuming and inaccurate numerical simulations?
“In addition, as shown in Figure 1, according to the reviewer’s suggestion, we used an optical measuring device and found that there is only a slight different from the existing results, e.g., in experiment 1the thickness reduction measured by micrometer was 0.561 and measured by an optical measuring devices 0.563, respectively. In other experiments, the results showed errors of less than 1%. Thus, we have chosen to stick to the existing method.” – it is not mentioned neither what kind of optical device was used nor how the thickness reduction was measured. If proper optical devices were available, why weren’t used? The above-mentioned comparison (micrometer vs. optical device), proving that micrometer is only 1% not as good as an expensive optical device (for this particular task, measuring thickness reduction for formed parts) would be extremely useful for other researchers.
In conclusion, I consider that none of the issues pointed by the reviewer were given satisfactory answers, so at this point I cannot recommend the paper to be accepted. Instead of bringing solid arguments backed up by solid literature references or by providing additional data, the authors just stick to their initial approaches.
Author Response
Dear Reviewer:
Firstly, we are so sorry that we couldn't give satisfactory answers to the Reviewer. At the same time, we are very grateful to have the opportunity to get the reviewer's suggestions again. This time, we have studied your comments carefully and have made corrections above. We sincerely hope that we will get a positive reply from you.
Please see the attachment for the cover letter.
Once again, thank you very much for your comments and suggestions.

Reviewer 3 Report
Thank you for addressing the comments.
However the representation of force is not accurate and must be improved. In SPIF one should consider compressive (vertical) and lateral (radial) components. In your graphs, such information is not provided. In this sense, please provide; 1) a ref. coordinate system and 2) the value of vertical and lateral components. You will see for future works that the optimization of each component individually may provide you really good results in terms of surface roughness and improved formability.
Author Response

(The authors gave the same response as above.)

Round 3
Reviewer 2 Report
At this stage of review, the authors have answered to all questions posed by the reviewer in a soundly and convincing way.
Issues which are still debatable are backed-up by literature references.
The processing toolpaths were properly described in the paper.
Measurements were performed by proper devices.
At this point, in my opinion, the paper is worthwhile to be published in the journal.
Author Response
Dear Reviewer:
Thank you for your letter and the reviewer’s comments concerning our manuscript entitled “RSM and BPNN modeling in incremental sheet forming process for AA5052 sheet: Multi-objective optimization using genetic algorithm” (Manuscript ID: metals-838932). We are very honored to have this exchange with the Reviewer. Thank you very much for your valuable suggestions, and your advice will have a significant impact on our future research.
Once again, thank you very much for your comments and suggestions.
Reviewer 3 Report
Reference 7 is outdated for a reference state of art. More recent ones can be found. A more recent one is "Single point incremental forming: An assessment of the progress and technology trends from 2005 to 2015", Journal of Manufacturing Processes 27, 37-62
Author Response
Dear Reviewer:
Thank you for your letter and the reviewer’s comments concerning our manuscript entitled “RSM and BPNN modeling in incremental sheet forming process for AA5052 sheet: Multi-objective optimization using genetic algorithm” (Manuscript ID: metals-838932). Those comments are all valuable and helpful for revising and improving our paper, as well as an important and significant guidance for our research.
Please see the attachment for the cover letter.
Once again, thank you very much for your comments and suggestions.
